# Analysis of the Glucose-Dependent Transcriptome in Murine Hypothalamic Cells

**DOI:** 10.3390/cells11040639

**Published:** 2022-02-11

**Authors:** Leonhard Webert, Dennis Faro, Sarah Zeitlmayr, Thomas Gudermann, Andreas Breit

**Affiliations:** Walther Straub Institute of Pharmacology and Toxicology, Medical Faculty, LMU Munich, Goethestrasse 33, 80336 Munich, Germany; leonhard.webert@online.de (L.W.); faro.dennis@gmail.com (D.F.); s.zeitlmayr@campus.lmu.de (S.Z.); thomas.gudermann@lrz.uni-muenchen.de (T.G.)

**Keywords:** glucose, hypothalamus, RNA-seq, cholesterol, insulin-like growth factor, noradrenalin, serum response element, cAMP

## Abstract

Glucose provides vital energy for cells and contributes to gene expression. The hypothalamus is key for metabolic homeostasis, but effects of glucose on hypothalamic gene expression have not yet been investigated in detail. Thus, herein, we monitored the glucose-dependent transcriptome in murine hypothalamic mHypoA-2/10 cells by total RNA-seq analysis. A total of 831 genes were up- and 1390 genes were downregulated by at least 50%. Key genes involved in the cholesterol biosynthesis pathway were upregulated, and total cellular cholesterol levels were significantly increased by glucose. Analysis of single genes involved in fundamental cellular signaling processes also suggested a significant impact of glucose. Thus, we chose ≈100 genes involved in signaling and validated the effects of glucose on mRNA levels by qRT-PCR. We identified *Gnai1*–*3*, *Adyc6*, *Irs1*, *Igfr1*, *Hras*, and *Elk3* as new glucose-dependent genes. In line with this, cAMP measurements revealed enhanced noradrenalin-induced cAMP levels, and reporter gene assays elevated activity of the insulin-like growth factor at higher glucose levels. Key data of our studies were confirmed in a second hypothalamic cell line. Thus, our findings link extra cellular glucose levels with hypothalamic lipid synthesis and pivotal intracellular signaling processes, which might be of particular interest in situations of *continuously* increased glucose levels.

## 1. Introduction

Glucose is a universal source of energy and a key player in diabetes and obesity. Although plasma glucose levels are kept rather constant at 5.50 mM after fasting, they can reach up to 10 mM after food intake, vary during the day and increasing after fasting even to 11 mM in diabetic patients [1,2,3,4]. Brain glucose levels directly correlate with plasma glucose levels, but depending on the specific brain region, may be 10 to 15 times lower than plasma levels [1,5,6]. Thus, the brain is also exposed to dynamic changes of extracellular glucose concentrations after food uptake and confronted with increased glucose levels in diabetic patients [1,7,8,9,10,11,12]. Furthermore, drugs such as heroin induce rapid and significant spikes in brain glucose levels [13], further highlighting the dynamics of brain glucose levels.

The hypothalamus is part of the hypothalamic–pituitary–thyroid (HPT) axis, and thus of prime importance for the central regulation of appetite and energy metabolism [14,15]. Dysregulation of the HPT axis has frequently been implicated in severe metabolic disorders including diabetes mellitus (DM) [16,17,18,19]. Signaling via receptor tyrosine kinases (RTK) such as the insulin-like growth factor receptor (IGFR) or G protein-coupled receptors (GPCR), such as β-adrenergic receptors, are important elements in the regulation of HPT activity [20,21,22,23,24,25]. Effects of altered extracellular glucose levels on signaling induced by both hormones in hypothalamic cells have not been systematically investigated thus far.

Besides the universal role of glucose as an energy carrier, glucose-derived ATP is an important regulator of intracellular signaling. In analogy to pancreatic β-cells, ATP inhibits K_ATP_ channels and thus depolarizes hypothalamic neurons [26,27,28,29,30]. Furthermore, ATP dampens AMP-dependent kinase (AMPK) activity in hypothalamic neurons, thereby relieving CRTC-2 from tonic phosphorylation and inhibition by AMPK [31]. AMPK activity inversely correlates with the activity of extracellular-regulated kinases-1/2 (ERK-1/2) in many cell types including hypothalamic cells [32,33]. Thus, glucose-induced AMPK inhibition leads to ERK-1/2 activation. Because ERK-1/2 are key to the activation of serum response factors (SRF) and ternary complex factors (TCF), increased glucose concentrations may lead to the activation of SRF/TCF-dependent gene expression via inhibition of AMPK. Finally, nuclear translocation of members of the transcription factor family max-like protein X is promoted by glucose-6-phosphate, another product of cytosolic glucose metabolism [34]. Thus, glucose per se has significant potential to serve as a key regulator of gene expression, as indicated already in liver cells and pancreatic islets [35,36]. However, no data about the glucose-dependent transcriptome in hypothalamic cells are available yet.

We have previously studied murine hypothalamic mHypoA-2/10 cells, establishing that changes between hypoglycemic extracellular glucose concentrations and high normoglycemic conditions (0.1 to 2.5 mM) regulate the expression of pro-thyrotropin-releasing hormone via inhibition of AMPK [37]. In this manuscript, we report the analysis of the entire glucose-dependent transcriptome in mHypoA-2/10 cells. We found that 831 genes were up- and 1390 genes were downregulated by at least 50%. The cholesterol biosynthesis pathway was most significantly affected on multiple levels, indicating regulatory effects of glucose on hypothalamic lipid synthesis. Further, qRT-PCR analysis of ≈100 genes involved in cellular signaling revealed a list of 15 genes including G proteins (*Gnai1*, *Gnai2*, *Gnai3*), adenylyl cyclase (*Adcy6*), and proteins involved in IGF signaling (*Irs1*, *Igfr1*, *Hras*, and *Elk3*) to be upregulated by glucose. Thus, we report thus far unappreciated effects of extracellular glucose levels on RTK and GPCR signaling in hypothalamic cells. These findings might be of importance in understanding the functional consequences of physiologically occurring alterations of extracellular glucose concentrations as well as *pathologically* increased glucose levels in diabetic individuals.

## 2. Materials and Methods

### 2.1. Materials

IGF-1, EGF, FSK, and noradrenalin were from Sigma-Aldrich (Taufkrichen, Germany). The SRF/TCF-dependent reporter was from Dr. Susanne Mühlich, Walter-Straub-Institute, LMU, München, Germany. Oligonucleotids (primer) were from Metabion, Martinsried, Germany. [2,8-^3^H]-adenine was from PerkinElmer (Boston, MA, USA). The anti-SDHA (ab14715) and the anti-IRS (ab131487) antibodies were from Abcam. The anti-SREBP-1 (sc-13551), anti-SREBP-2 (sc-13552), and the anti-ERK1 D2 (sc-1647) antibodies were from Santa Cruz. The anti-ADCY6 (PA5-101284) antibody was from Thermo Fisher Scientific.

### 2.2. Cell Culture and Transfection

The adult mouse hypothalamic cell line mHypoA-2/10 (ordering number Clu-176) was from Cedarlane. mHypoA-2/10 express a wide array of hypothalamic markers [38]. We have recently reported that these cells behave similar to *thyreoliberin*-positive neurons of the *paraventricular nucleus* and that they respond to a wide range of hormones such as melanocortins, serotonin, catecholamines, and IGF [20,37,39,40,41]. GT1-7 cells were kindly provided from Dr. Weiner (University of California, San Francisco, CA, USA) and behave like GnRH-secreting cells of the hypothalamus [42,43]. All cells were cultured in DMEM with 10% FBS, 25 mM glucose, 2 mM L-glutamine, and penicillin/streptomycin at 37 °C and 5% CO_2_ [38]. mHypoA-2/10 cells stably expressing the SRF/TCF reporter were previously published [20]. In order to downregulate CRTC-2 expression, we recently established the following protocol [37]. A cocktail of three specific siRNAs (sc-45833) or a control siRNA (sc-37007) from Santa Cruz was transfected into mHypoA-2/10 cells via electroporation using the Neon^®^ transfection system from Invitrogen (Schwerte, Germany) according to the manufacturer’s protocol. Briefly, for one pulse, 500,000 cells together with 40 nM of the corresponding siRNA were challenged with 1450 V for 30 ms.

### 2.3. Total RNA-seq

Isolation of total RNA, RNA-seq experiments, and gene expression analysis were performed by IMGM Laboratories GmbH, Planegg, Germany. After stimulation of the cells in house, the reaction was stopped by adding RNA stabilizer (RNAlater from LifeTechnologies, Taufkirchen, Germany). Samples were frozen in liquid nitrogen and stored at −80 °C. After transportation on dry ice, lysates were thawed, and total RNA was extracted using the RNeasy Mini Kit (Qiagen, Hilden, Germany). TruSeq^®^stranded mRNA HT kit (Illumina, Baltimore, MD, USA) was used for library preparation according to the manufacturer’s protocol. Amplified cDNA was purified using a solid phase reversible immobilization (SPRI) paramagnetic bead-based technology (AMPure XP beads, Beckham Coulter, Krefeld, Germany) with a bead to DNA ratio of 1 to 1. Quality control of each sample was carried out using 2100 Bioanalyzer DNA 1000 LapChip (Agilent Technologies, Waldbronn, Germany), and cDNA quantification was determined using a Qubit^®^ ds DNA assay kit (Thermo Fisher Scientific, Waltham, MA, USA). cDNA was denaturated by NaOH, and sequencing was performed with a 1% PhiX v3 control library spike (Illumina, Baltimore, MD, USA) in a NextSeq^®^ 500 sequencing system (Illumina, Baltimore, MD, USA). Primary image processing was performed on the NextSeq^®^ 500 instrument using Real Time Analysis 2.4.11 Software (RTA). Primary data analysis was performed using the bcl2fastq 2.15.04 software package. The Illumina Sequence Analysis Viewer (SAV) 2.1.8 was used for imaging and evaluation of the sequencing run performance. The CLC Genomics Workbench 9.5.3 (CLC bio, a Qiagen, Hilden, Germany) was used for in-depth analysis of differential gene expression. Image and signal processing, such as imaging and base calling, was carried out by the Illumina NextSeq^®^ 500 inherited NCS and RTA software packages through applying the FastQ only processing pipeline. Data obtained were mapped against the mouse reference genome (GRCm38.p4) downloaded from NCBI as reference sequence. The expression values were processed to RPKM, a normalized measure of relative abundance of transcripts. It is defined as reads per kilobase of exon model per million mapped reads. For the analysis of differential gene expression, the three data sets of low glucose were compared to three data sets of cells kept at high glucose. The method of Benjamini and Hochberg was used to control for the false discovery rate (FDR). Bonferroni-corrected *p*-values were also calculated. Functional analysis was performed using the Ingenuity Pathway Analysis (IPA) software, build version: 463341M, content version: 42012434.

### 2.4. Quantitative RT-PCR (qRT-PCR)

Serum-starved mHypoA-2/10 cells were either kept at low- or high-glucose concentrations as described below. Stimulation was terminated by rapid cooling on ice, and total RNA was isolated using the Trizol^®^ reagent (Invitrogen, Darmstadt, Germany) according to the manufacturer’s instructions. First strand synthesis was carried out with oligo(dT)_18_ primer using 2 µg of total RNA and the RevertAid^™^ H Minus First Strand cDNA Synthesis Kit (Fermentas, Sankt-Leon Roth, Germany). qRT-PCR was conducted using the LightCycler^®^ 480 SybrGreen I Master Mix kappa (Roche, Mannheim, Germany), intron-spanning primer pairs at a final concentration of 1 µM each, and 0,08 µL of the first strand synthesis reaction in a LightCycler^®^ 480 II (Roche) using the following conditions: initial denaturation for 2 min at 94 °C, 55 cycles of 94 °C for 10 s, 55 °C for 10 s, and 72 °C for 10 s. Primer sequences are provided in Appendix A. Crossing points (Cp) were determined by the software supplied with the LightCycler^®^ 480, and data were analyzed by the ΔΔCp method (2^-((target gene–ref. gene)^_high glucose_^-(target gene–ref. gene))^_low glucose_).

### 2.5. Glucose Stimulation

Average brain glucose concentrations in rodents range between 1.3 and 2.5 mM. In living human beings, an average brain glucose level of 1.0 ± 0.1 mM is detected [5]. Hypothalamic glucose concentrations are approximately half the average of brain glucose concentrations [9,26,28,44,45,46]. In order to analyze the entire glucose-dependent transcriptome, we aimed at culturing mHypoA-2/10 cells at the extreme ends of the physiological possible glucose concentration. Thus, we kept the cells under hypoglycemic conditions and compared them with 2.5 mM glucose, which might represent the highest hypothalamic glucose level under normoglycemic conditions. Two distinct protocols were used. For RNA analysis, cells were first starved for 24 h with 0.1 mM glucose in DMEM with 1 mM pyruvate in the absence of FBS. Afterwards, one pool of cells was kept at 0 mM, and the corresponding pool at 2.5 mM glucose for 3 h (protocol 1). For reporter, cAMP, whole-cell ELISA or cholesterol assays cells were kept for 24 h at 0.1 or 2.5 mM glucose in the absence of FBS (protocol 2).

### 2.6. Reporter Gene Assay

Cells were lysed in 100–200 µL of lysis buffer (25 mM Tris/HCl (pH 7.4), 4 mM EGTA, 8 mM MgCl_2_, 1 mM DTT, and 1% Triton-X-100), and a volume of 90–180 µL was transferred to white-bottomed, 96-well plates. Luciferase activity was measured after automatically injecting the luciferase substrate (20–40 µL) using a FLUOstar^®^ Omega plate reader (BMG, Offenburg, Germany). Total light emission was detected every second for 10 s post-injection, and average emission between 2 and 10 s was calculated.

### 2.7. Whole-Cell ELISA

Whole-cell ELISA assays were performed as described previously with slight modifications [37,39,47]. Electroporated cells were seeded in 24-well plates (≈15,000/well) two days prior to the experiment. The next day, cells were kept at either 0.1 or 2.5 mM glucose for 24 h. Total SREBP-1, SREBP-2, IRS-1, or AC-6 expression was then analyzed with the according antibody. The anti-SREBP-1 (sc-13551) and the anti-SREBP-2 (sc-13552) antibody was from Santa Cruz. The anti-AC6 (PA5-101284) antibody was from Thermo Fisher Scientific, and the anti-IRS (ab131487) antibody was from Abcam. After fixation for 15 min with 4% PFA, cells were permeabilized with ice-cold methanol/acetone (50:50) for 5 min. Next, cells were washed with PBS and then incubated with 1.0% of BSA in PBS for 15 min at RT to block unspecific binding sites. After an additional washing steps, cells were incubated for 30 min at RT with the corresponding first antibody (1:2000) in PBS/BSA. As a control, a pool of cells was incubated in PBS/BSA without first antibody. Next, antibodies were removed, and cells were washed and incubated with the corresponding HRP-conjugated secondary antibody (1:4000) in PBS/BSA for 30 min at RT. After several washing steps, 200 µL of the 1-Step^TM^-Ultra-TMB-ELISA substrate (ThermoScientific, Rockford, IL, USA) was added to the cells and incubated for 15 min at RT. The reaction was stopped by adding 50 µL of 1 M H_2_SO_4_, 200 µL was transferred to 96-well plates with clear bottom, and OD_450_ was measured in a FLUOstar^®^ Omega plate reader. Background values of the control cells (without first antibody) were subtracted.

### 2.8. Cholesterol Detection

In order to detect cholesterol levels, we used the cholesterol ester-glo assay^®^ from Promega (J3190) according to the manufacturer’s protocol. Dependent on the amount of cholesterol present, cholesterol dehydrogenase produces NADH. This NADH is used by a reductase to convert a non-luminescent reductase substrate into luciferin, which can be detected in the presence of the ultra-glo^TM^ luciferase, ATP, and Mg^2+^ on the basis of its bioluminescence. Thus, cholesterol levels directly correlate with the bioluminescence measured. Because glucose might affect NADH or ATP levels in cells, we avoided direct cholesterol measurements in the cells. We chose to extract cellular cholesterol with cyclodextrins. In detail, we first cultured cells with 0.1 or 2.5 mM glucose for 24 h in 96-well plates (≈10,000 cells per well) and then extracted cholesterol with 20 mM β-cyclodextrin in 100 µL DMEM with 0.1 mM glucose and 2% BSA for 4 h. A total of 20 µL of these supernatants was used to detect cholesterol levels independently from NADH or ATP levels in the cells. Cholesterol standards were used to normalize luminescence signals to the cholesterol concentration.

### 2.9. cAMP Accumulation

To determine agonist-induced cAMP accumulation, we seeded 100,000 cells in 12-well dishes 48 h prior to the experiment and labelled them in serum-free DMEM containing 1 µCi/mL of [^3^H]-adenine as described above. Cells were stimulated for 20 min in DMEM containing 1 mM 3-isobutyl-1-methylxantine along with noradrenalin or FSK. The reaction was terminated by removing the medium and adding ice-cold 5% trichloroacetic acid to the cells. [^3^H]-ATP and [^3^H]-cAMP were then purified by sequential chromatography (dowex-resin/aluminium oxide columns). Radioactivity was measured by scintillation counting using a WinSpectral 1414 (PerkinElmer, Rodgau, Germany).

### 2.10. Quantification and Statistical Analysis

Values represent the mean ± SD of three to six independent experiments. Statistical analysis was performed using one or two-sample Student’s *t*-test or two-way ANOVA followed by Sidak’s or Tuckey’s post-test using the GraphPad prism software 9.1. One symbol indicates a *p*-value of <0.05, two of <0.01, and three of <0.001.

## 3. Results

### 3.1. Cholesterol Biosynthesis Was the Major Target of Glucose-Dependent Gene Expression

We recently introduced protocols for glucose stimulation of mHypoA-2/10 cells (Section 2.5). On the basis of these protocols, we initially adapted cells to low extracellular glucose concentrations (0.1 mM) for 24 h in the absence of fetal bovine serum (FBS) [37]. Afterwards, glucose either was withdrawn (no glucose) or increased to 2.5 mM (glucose) for 3 h (protocol 1, Section 2.5). Pyruvate (1 mM) was constantly present to ensure equal mitochondrial ATP levels. This protocol led to enhanced activity of a CREB/CRTC2-dependent reporter gene but did not affect activity of signal transducers and activators of transcription or of nuclear factors-activated in T cells [37].

Here, we took advantage of this protocol and performed total RNA sequencing (RNA-seq) under both conditions in order to determine the glucose-dependent transcriptome in murine hypothalamic cells. Three samples were analyzed for each condition. Sequencing depth for low glucose was 1127 ± 131 M and for high glucose 1153 ± 227 M total reads per sample, which indicates a sequencing coverage of 11.1 ± 1.2 and 11.3 ± 1.9, respectively. The resulting reads per kilobase of transcript per million mapped reads (RPKM) were mapped against the *Mus musculus* reference genome (GRCm38.p4 down-loaded from NCBI) and analyzed for differential gene expression. When a fold change ǀFCǀ of ≥2 in mRNA expression was applied as a cut-off, 641 genes were up- and 766 downregulated (FDR-corrected *p*-value of ≤0.05) by glucose (a list of all changed genes is shown in Appendix A). Thus, in relation to the entire mouse genome (≈45,000 genes), ≈3% of the transcriptome was found to be glucose-dependent. When a cut-off for total (glucose-independent) mRNA expression was applied (RPKM of ≥1 for at least one condition), 11,062 genes remained (Figure 1). A total of 187 of these genes were up- and 480 genes were downregulated by a ǀFCǀ of ≥2, and 831 or 1390 by a ǀFCǀ of ≥1.5, representing 6.0 or 20%, respectively, of the transcriptome with a RPKM ≥ 1 (Figure 1). Hence, total RNA-seq analysis revealed a significant impact of glucose on the transcriptome of hypothalamic cells. Canonical signaling pathway analysis revealed the cholesterol biosynthesis gene network to be most profoundly affected by glucose (Figure 2). As shown in Figure 3A,B, nine key genes involved in cholesterol and four in fatty acid biosynthesis were significantly upregulated by glucose. SREBP are encoded by the *Srebf1* and *Srebf2* genes and are key regulators of cholesterol biosynthesis [48]. Glucose has been shown to enhance *Srebf1* and *Srebf2* expression in renal NRK-52E cells, but conversely to inhibit *Srebf1* in Schwann cells [49,50]. RNA-seq data obtained with mHypoA-2/10 cells showed that both *Srebf* genes were upregulated (Figure 3C), portending a major role of glucose as a regulator of cholesterol and fatty acid levels by dynamic regulation of *Srebf* expression.

It has been shown that the CREB co-activator CRTC-2 regulates SREBP-1 expression in liver cells [51]. We have recently reported that glucose activates CRTC-2 in mHypoA-2/10 cells [37]. Thus, CRTC-2 activation appeared as a likely mechanism for glucose-induced SREBP expression. We used a siRNA cocktail against CRTC-2 and detected the effects of glucose on SREBP expression by a whole-cell ELISA approach (Figure 4A). Glucose significantly enhanced SREBP protein expression (3.2 ± 0.3 for SREBP-1 and 2.4 ± 0.2-fold over basal for SREBP-2) after transfection of a control siRNA. In the presence of the specific CRTC-2 siRNA, effects of glucose on the expression of both SREBP isoforms were almost completely abolished, indicating a major role for CRTC-2 in glucose-induced SREBP expression. To validate finally the effect of glucose on cholesterol synthesis and the role of CRTC-2 in this context, we determined cholesterol levels in mHypoA-2/10 cells after a 24 h treatment with 0.1 or 2.5 mM glucose. Cyclodextrins have a hydrophobic core that is roughly the size of cholesterol. Thus, in the presence of β-cyclodextrin, cholesterols are extracted from cells. Glucose-induced cholesterol synthesis should therefore lead to enhanced β-cyclodextrin-promoted cholesterol release into the supernatant of mHypoA-2/10 cells. Indeed, the cholesterol levels in the supernatant of cells transfected with the control siRNA rose from 37.8 ± 6.5 to 58.7 ± 3.9 µM in the presence of high glucose concentrations (Figure 4B). This increase was significantly lower (33.4 ± 3.4 to 43.1 ± 5.1 µM) in the supernatant of cells transfected with the siRNA against CRTC-2 (Figure 4B). Hence, our data clearly indicate that glucose regulates cholesterol levels of mHypoA-2/10 cells, most likely via CRTC-2-mediated expression of SREBP-1 and -2. In order to investigate whether these effects are restricted to one particular hypothalamic cell line, we took advantage of GT1-7 cells, a distinct murine hypothalamic cell line [52]. Similar to mHypoA-2/10 cells, glucose enhanced β-cyclodextrin-induced cholesterol release in GT1-7 cells (Figure 4C). Hence, effects of glucose on cholesterol synthesis appeared as a rather common feature among hypothalamic cell lines.

### 3.2. Glucose-Dependent Gene Expression Altered Signaling Pathways of RTK and GPCR

After assessing the effects of glucose on entire signaling networks, we subsequently focused on single genes that are involved in intracellular signaling cascades. We found that mRNA levels of monomeric and heterotrimeric G proteins are modulated by glucose, as are corresponding signaling molecules such as GTPase-activating protein, GTP exchange factors, and transcription factors. Specifically, we found 31 of these genes to be up- and 45 to be downregulated by glucose (Figure 5). Further, glucose apparently exerted rather specific effects on certain subtypes within the same gene family. For example, the transcription factor *Elk3* but not *Elk1* was significantly induced (Figure 5, middle panel). Similarly, expression of *Irs1* but not of *Irs2* was enhanced (Figure 5, lower panel). With regard to α-subunits of heterotrimeric G proteins, *Gnai1*, *Gnai2*, *Gnai3*, and *Gna11* expression was enhanced; *Gnaq* and *Gna12* expression was unchanged; while *Gna13* expression was reduced by glucose (Figure 5). Next, we aimed at confirming the effects of glucose on the mRNA levels by qRT-PCR. To this end, we created specific primer pairs for 31 up- and for 45 downregulated genes (Appendix A). As a control, we also included 23 genes for which expression was unchanged. In addition, we also screened the RNA-seq data set for potential reference genes not affected by glucose. We found that mRNA levels of succinate dehydrogenase (*Sdha*) and TATA-box-binding protein (Tbp) were glucose-independent (Appendix A). Further, Sdha expression exceeded that of Tbp by a factor of ≈4, thus allowing for controls at different expression levels. We stimulated cells according to the protocol used for the RNA-seq approach and analyzed mRNA levels of Sdha and Tbp with different amounts of cDNA by qRT-PCR. Both genes showed almost perfect linear correlation between crossing points (Cp-values) and cDNA amount (Appendix A). Furthermore, over the entire range of cDNA concentrations, glucose had no effect on the mRNA expression of either Sdha or Tbp. Thus, we generated three independent pairs of mRNAs from cells +/− glucose stimulation and performed qRT-PCR with the abovementioned 99 genes and analyzed their mRNA expression normalized to Sdha or Tbp. The heat map in Figure 6 combines the glucose-induced changes obtained in the RNA-seq experiment shown in Figure 5 with the qRT-PCR data obtained. The entire set of qRT-PCR data is shown in Appendix A. Some candidates derived from the RNA-seq experiment such as Igf2r, Kras, Elk4, Rab7b, and Arhgef19 were insensitive to glucose in the qRT-PCR. Others, such as Srf, Crtc2, and Arhgef39 exhibited opposite effects when RNA-seq was compared to qRT-PCR data. However, for 65 of the 99 genes analyzed, the effect of glucose on mRNA levels observed in RNA-seq was confirmed by the qRT-PCR experiments for at least one reference gene. Of note, the heat map only indicates absolute changes, but does not provide any information about the significance of these changes. Thus, we performed a rigorous statistical analysis based on three parameters: (1) Bonferroni-corrected *p*-values of the RNA-seq data, (2) ANOVA/Sidak’s *p*-values of the qRT-PCR experiment using *ΔΔ*Cp on the basis of Sdha or Tbp, and (3) two-sample t-test-based *p*-values of the qRT-PCR experiment using *ΔΔ*Cp on the basis of Sdha or Tbp. Genes, for which glucose-dependent mRNA levels reached statistical significance (*p*-value of ≤ 0.05) in all three categories and with both reference genes, are listed in Table 1.

*Irs1* appeared as the most profoundly glucose-induced gene (72 ± 26%). Interestingly, *Irs2* expression was unchanged by altered glucose concentrations (Figure 5 and Figure 6, Appendix A). Considering the RNA-seq data, it appeared that glucose-induced *Irs1* expression increased total RPMKs of both *Irs* from 9.8 to 11.6 (Figure 7A). Further, although glucose selectively increased *Irs1* mRNA levels by almost twofold, the total expression level of *Irs2* was still significantly higher, indicating that *Irs2* remained the most abundant *Irs* subtype upon glucose treatment in mHypoA-2/10 cells (Figure 7A). Finally, RNA-seq data also suggested a tendency of glucose-induced *Irs2* depression. Thus, we cultivated mHypoA-2/10 cells for 24 h either with 0.1 or with 2.5 mM glucose (protocol 2) and analyzed mRNA levels of *Irs1* and *Irs2* again by qRT-PCR (Figure 7B). We could confirm a significant increase of *Irs1* expression by 48 ± 15% and observed a significant glucose-induced decrease of *Irs2* expression by 12 ± 6% in the same cells. Thus, glucose induced an increase in total *Irs* expression and a shift from *Irs2* to *Irs1* in hypothalamic cells. Of note, the *Igfr1* gene was also induced by glucose (Figure 5 and Figure 6, Table 1), suggesting that glucose modulates IGF signaling by three distinct mechanisms: increase of total *Irs* and *Igfr1* expression and a changed expression ratio of *Irs2* towards *Irs1*. In order to validate this hypothesis, we analyzed effects of glucose on the protein level of IRS-1 by ELISA (Figure 7C). We observed that glucose induced expression of the IRS-1 protein after 24 h by 3.1 ± 0.9-fold over basal. Downregulation of CRTC-2 with siRNA blocked glucose-induced IRS-1 expression almost completely (Figure 7C), indicating that the CREB co-activator is involved.

We next validated the consequences of glucose-induced IRS-1 expression on the function of RTK by expressing a luciferase reporter gene containing binding sites for SRF/TCF in mHypoA-2/10 cells [53]. We incubated these cells either with 0.1 or 2.5 mM glucose for 24 h and analyzed IGF- or epidermal growth factor (EGF)-induced, SRF/TCF-dependent reporter activity. When analyzing basal reporter activity, we found an increase of 220 ± 73% under high-glucose condition (Figure 7D). This increase could be caused either by glucose-induced expression of *Hras* and/or *Elk3* shown in Table 1, or by the aforementioned glucose-promoted ERK-1/2 activation via AMPK inhibition [32,33]. Interestingly, IGF further enhanced SRF/TCF activity at high glucose by 170 ± 28% and only by 115 ± 15% at low glucose (Figure 7E). Thus, as expected from our expression analysis, glucose enhanced IGF signaling. Of note, EGF-mediated effects on SRF/TCF activity were glucose-independent (Figure 7E), which is in line with the inconsistent effects of glucose on the expression of EGF receptors or their adapter proteins GAB or GRB (Appendix A) and the glucose-independent expression of the *Sos1* and *Sos2* genes (Figure 5, Appendix A). Thus, extracellular glucose concentrations exhibited rather selective effects on IGF, but not on EGF signaling. Of note, this also appeared as a rather common feature of hypothalamic cells since glucose treatment also affected IGF but not EGF signaling in GT1-7 cells (Figure 7F).

Our list of glucose-induced genes also includes three α-subunits of the *Gnai* family and *Adcy6*, suggesting effects of glucose on cAMP signaling. An increase of total gnai expression by glucose implies decreased cytosolic cAMP levels. In contrast, increased expression of *Adcy6* should result in enhanced cAMP signaling. Of note, *Adcy6* is the predominant adenylyl cyclase (AC) in mHypoA-2/10 cells (Figure 8A). Thus, the relative increase of *Adcy6* mRNA (44 ± 11%, Table 1) has robust effects on the total amount of *Adcy* expression (total *Adcy* RPMK increased from 16.9 at basal to 23.6 with glucose). Further, analysis of the glucose-dependent protein level (Figure 8B) revealed enhanced AC-6 expression at higher glucose concentrations. This effect was also diminished after transfection of the CRTC-2 siRNA, albeit not completely (Figure 8B). In order to determine the effects of glucose on cAMP levels, we incubated cells for 24 h either with 0.1 or with 2.5 mM glucose in the presence of [^3^H]-adenine and measured basal, noradrenalin (NA)-, and forskolin (FSK)-induced cAMP accumulation afterwards. NA activates β_2_- and β_3_-AR in mHypoA-2/10 cells, and FSK is a direct activator of AC [41]. Using sequential chromatography, we first tested whether glucose would affect the conversion of [^3^H]-adenine to [^3^H]-ATP. Because ATP is the substrate of the reaction catalyzed by AC, altered labelling efficiency due to different cytosolic glucose concentrations may theoretically affect the enzymatic reaction independently from AC activity. As shown in Figure 8C, [^3^H]-ATP levels were not significantly affected by extracellular glucose levels, indicating that the glucose stimulation protocol did not affect labelling of the cells with [^3^H]-adenine. When the product of the AC reaction was analyzed, slightly decreased basal [^3^H]-cAMP levels were detected at high glucose concentrations (Figure 8D). When NA- and FSK-induced cAMP accumulation were determined, significantly increased x-fold over basal signals were obtained at high glucose concentrations (Figure 8E). Interestingly, when x-fold over basal signals obtained at 0.1 mM glucose were set to 100%, high glucose levels increased NA-induced cAMP accumulation by 123 ± 46% and FSK-promoted signals by 20 ± 28%, indicating selective effects of glucose on NA-induced signaling (Figure 8F).

## 4. Discussion

Plasma and brain glucose levels vary under physiological and pathological conditions. Glucose has been established as a regulator of gene expression in several tissues, but glucose-dependent gene expression in hypothalamic cells has not yet been analyzed in detail [36,54,55,56,57]. Altered gene expression has been reported in the brain of diabetic rodents, but in vivo models can hardly differentiate between the effects of glucose or insulin on gene expression [58,59]. Herein, we analyzed glucose-dependent gene expression in vitro, independent of insulin using a hypothalamic cell line and a four-step strategy: (1) we determined the entire transcriptome of cells at low and high glucose levels by RNA-seq; (2) we selected candidates and re-evaluated their glucose-dependent mRNA levels by qRT-PCR; (3) we monitored protein expression levels by ELISA; and (4) we determined cholesterol levels, second messengers, or reporter gene activities at different glucose concentrations.

RNA-seq data indicated that depending on the cut-off of total gene expression and the amount of change, 700–2200 genes were glucose-dependent, corresponding to around 3 to 7% of the entire murine genome. Previous studies using either healthy or diabetic human/murine islets or skin cells obtained similar results [54,55,56,57].

The cholesterol biosynthesis pathway was most prominently induced. Glucose is fundamental for cholesterol synthesis because its degradation product pyruvate is converted to acetyl CoA, which serves as the precursor of cholesterol synthesis. Of note, we kept pyruvate levels constant at 1 mM, such that pyruvate deficiency did not occur under low glucose conditions. Thus, our data show that alterations of cytosolic products of glucose metabolism (most likely ATP) affect the expression of SREBP’s and of enzymes involved in the cholesterol synthesis pathway. In line with this notion, glucose enhanced cholesterol levels in two distinct hypothalamic cell lines. Thus, our data indicate a strong correlation between extracellular glucose levels and the cholesterol synthesis pathway in hypothalamic cells. AMPK is inhibited by ATP and inhibits CRTC-2 activity in hypothalamic cells [31,37]. Hence, CRTC-2 appeared as a promising candidate for glucose-induced SREBP expression, particularly when considering that CRTC-2 has been reported to induce SREBP-1 expression in liver cells [51]. In line with these data, downregulation of CRTC-2 expression by siRNA diminished the effects of glucose on the expression of both SREBP subtypes and on cholesterol synthesis. In addition to the positive effects of glucose on the cholesterol levels in mHypoA-2/10 cells, the insulin-induced gene-1 (*Insig1*) gene belongs to the 15 glucose-dependent genes revealed herein (Table 1). *Insig1* RNA levels were enhanced by 62 ± 23% at high glucose levels. INSIG-1 is the most profoundly insulin-induced protein in liver cells and functions as an inhibitor of SREBP activity and thus as a negative regulator of cholesterol synthesis [60,61]. Insulin-independent effects of glucose on *Insig1* expression have not yet been shown. Furthermore, induction of *Srebf* and *Insig1* in hypothalamic cells by glucose suggests effects on cholesterol synthesis on multiple cellular levels. DM is associated with alterations of the central nervous system including reduction of cholesterol synthesis in the brain [62,63]. Glucose-induced expression of enzymes involved in cholesterol synthesis indicated by our data and by others appears to contradict this notion, taking into account increased plasma glucose levels in diabetic individuals [64]. However, glucose-induced INSIG-1 expression may account for reduced cholesterol levels in DM. Glucose insensitivity or tolerance is a hallmark in the origin and progression of DM. Thus, although glucose initially induces cholesterol synthesis, this effect might diminish after sustained elevated glucose levels, finally resulting in reduced cholesterol levels. Fukui et al. recently demonstrated in GT1-7 cells that reduction of cellular cholesterol levels by 30% reduced IGF-induced ERK-1/2 phosphorylation [65]. These data are in accordance with increased IGF-induced SRF/TFC activation at high-glucose concentrations, as shown in Figure 6. Thus, in addition to the already mentioned effects of glucose on total *Irs* and *Igfr1* expression and the altered *Irs2* versus *Irs1* expression ratio, glucose might also affect IGF signaling via modulating cholesterol levels.

We also observed depression of *Gnai* genes and induction of the *Adcy6* gene. To the best of our knowledge, this is the first description of direct modulation of key proteins involved in cAMP production by glucose. Accordingly, when measuring glucose-dependence of cAMP accumulation, we found dual effects of glucose on cAMP signaling. Inhibition of basal cAMP levels could be attributed to enhanced expression of *Gnai* genes, whereas increased *Adcy6* expression may account for enhanced NA-promoted cAMP signaling. NA-induced cAMP signals were independent from pertussis toxin, an inhibitor of all G_i_ proteins in mHypoA-2/10 cells (data not shown), providing a possible explanation as to why enhanced G_i_ protein expression by glucose did not affect NA-mediated cAMP signaling. Glucose also enhanced cAMP accumulation induced by the direct AC activator FSK, but to a significantly smaller extent compared to NA (Figure 8F). Interestingly, in contrast to NA, pertussis toxin treatment affected FSK-induced cAMP accumulation (data not shown), suggesting that the involvement of G_i_ proteins in ligand-induced cAMP accumulation contributes to the glucose effect on this process. However, NA is a key activator of the HPT axes because it induces thyreoliberin release from the hypothalamus via cAMP production [22,41]. Thus, it is reasonable to assume that glucose-induced enhancement of NA signaling might activate the HPT axis and thus enhance metabolism. Dysregulation of this process in diabetic individuals may provide a new link between DM and altered metabolic activity.

Overall, we discovered thus far unappreciated effects of glucose on gene expression in murine hypothalamic cells, affecting cholesterol biosynthesis as well as IGF and NA signaling. Effects of glucose on cholesterol synthesis, SREBP-1 and -2, or IRS-1 expression were completely—but on AC-6 expression partially—dependent on CRTC-2, suggesting a key role of the CREB co-activator. This is in line with our previous study, showing that glucose enhances CRTC-2 activity by inhibition of AMPK [37]. However, at this point, we cannot exclude the participation of other factors, such as SRF/TCF or max-like protein X. Detailed promoter analysis of the 15 genes shown in Table 1 will be required in order to dissect the exact mechanism leading to glucose-induced gene regulation.

Lastly, we employed an in vitro model and rather pronounced changes of glucose concentrations. Therefore, at this point, we can only speculate to what extent our data can be extrapolated to the in vivo situation. Hence, it will be an enlightening task for future studies to validate which of the genes identified in hypothalamic cell models are, indeed, glucose-dependent in vivo and whether or not these changes contribute to the pathophysiology of DM.

## Figures and Tables

**Figure 1 cells-11-00639-f001:**
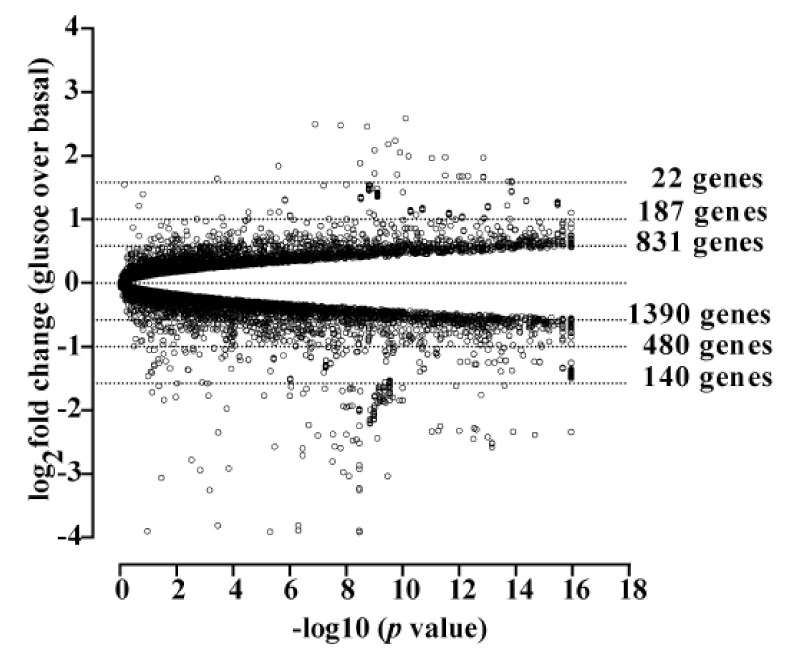
Volcano plot of RNA-seq data obtained with mHypoA-2/10 cells at low and high glucose levels. Cells were stimulated according to protocol 1 illustrated in Section 2.5. Only genes with RPKM of ≥1 for at least one condition (11,062 genes) are shown. The number of genes with a fold change of ǀFCǀ ≥ 3, ǀFCǀ ≥ 3, or ǀFCǀ ≥ 1.5 are given. For each condition, one single experiment performed in triplicate was analyzed as described under experimental procedures.

**Figure 2 cells-11-00639-f002:**
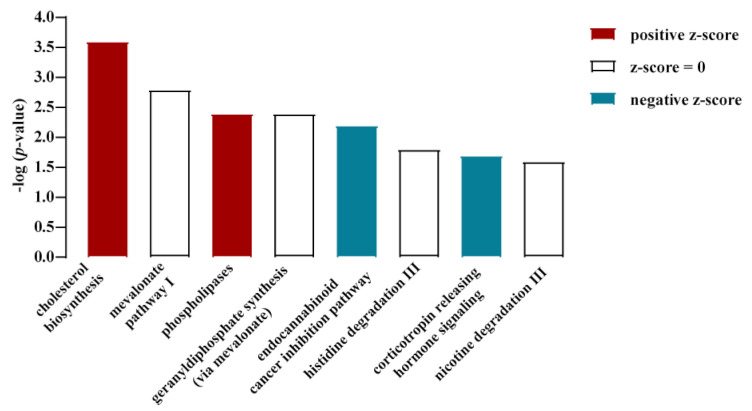
Canonical signaling pathway analysis of RNA-seq data obtained with mHypoA-2/10 cells at low and high glucose levels. For details, see Section 2.3.

**Figure 3 cells-11-00639-f003:**
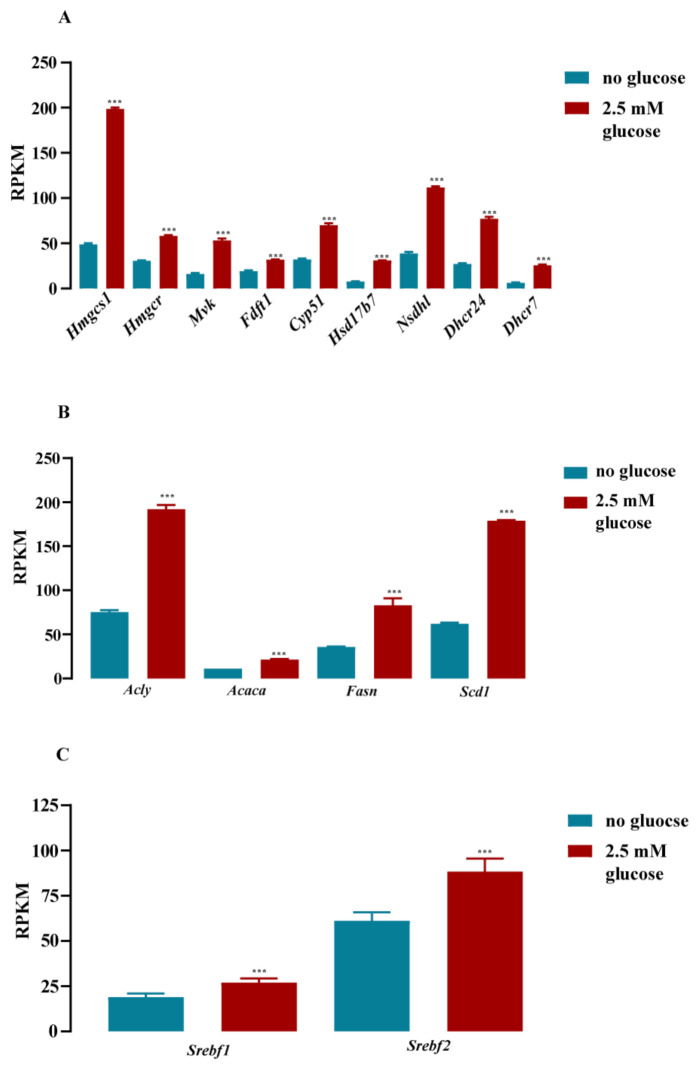
Glucose induced RNA expression of both Srebf genes and of key enzymes involved in cholesterol or fatty acid biosynthesis in mHypoA-2/10 cells. Cells were stimulated according to protocol 1 illustrated in Section 2.5 and RPKM determined by RNA-seq. For each condition, triplicates from one single experiment were analyzed as described under experimental procedures. In (**A**), data for 3-hydroxy-3-methylglutaryl-CoA synthase 1 (Hmgcs1), 3-hydroxy-3-methylglutaryl-CoA reductase (Hmgcr), mevalonate kinase (Mvk), farnesyl-diphosphate farnesyltransferase-1 (Fdft1), lanosterol 14α-demethylase (Cyp51), hydroxysteroid 17-β dehydrogenase-7 (Hsd17b7), NAD(P)-dependent steroid dehydrogenase-like (Nsdhl), 24-dehydrocholesterol reductase (Dhcr24), and 7-dehydrocholesterol reductase (Dhcr7) are shown. In (**B**), data for the ATP citrate lyase (Acly), acetly-Coa carboxylase-α (Acaca), fattay acid synthase (Fasn), and stearoyl-Coa desaturase (Scd1) are presented. In (**C**), data for the sterol regulatory element-binding protein gene-1 (Srebf1) and -2 (Srebf2) are presented. Data are expressed as the mean ± S.E.M. of one single experiment (N = 1) performed in triplicate. Asterisks indicate significant differences calculated on the basis of Bonferroni-corrected *p*-values of the entire data set obtained by RNA-seq. *** *p* < 0.001.

**Figure 4 cells-11-00639-f004:**
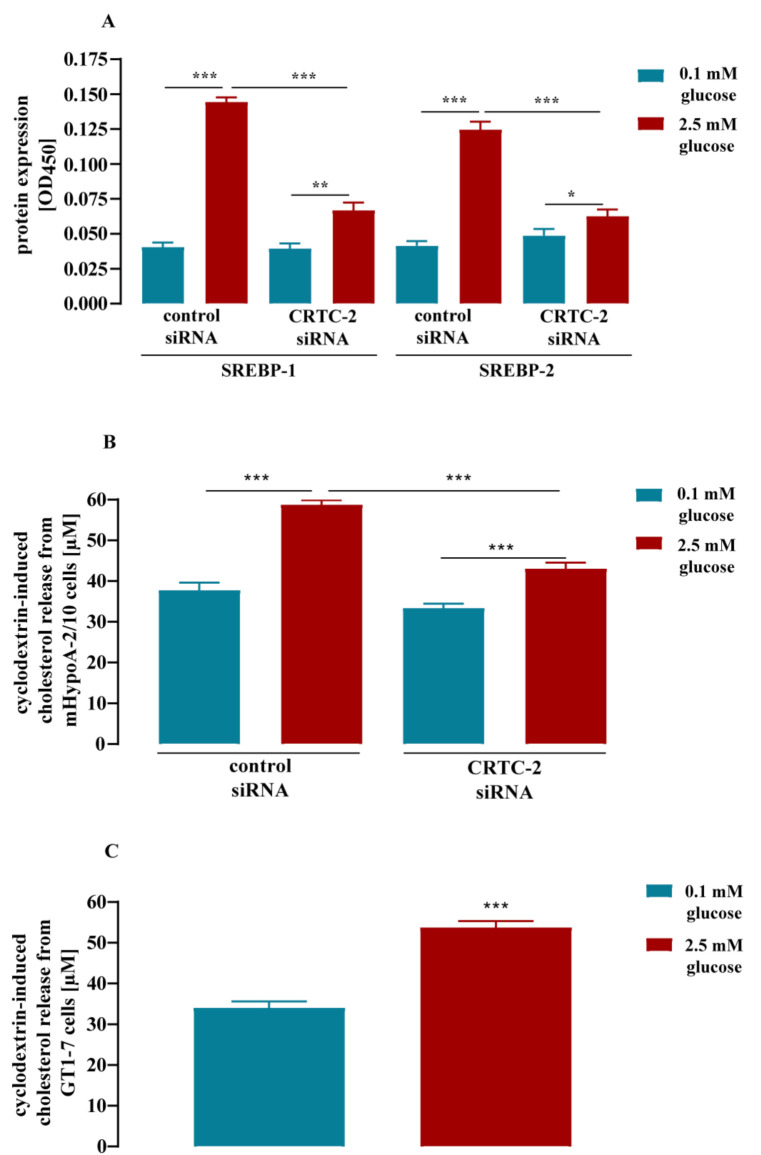
Glucose-induced SREBP protein expression and cholesterol synthesis via CRTC-2 in mHypoA-2/10 cells. Cells were stimulated according to protocol 2 illustrated in Section 2.5. (**A**,**B**) mHypoA-2/10 cells were transfected with a control or a specific siRNA against CRTC-2. (**A**) SREBP-1 or -2 expression was analyzed by whole-cell ELISA. (**B**) Cholesterol levels in the supernatant were detected after incubation of the cells with 20 mM β-cyclodextrin for 4 h. (**C**) Cholesterol levels were determined in the supernatant of GT1-7 cells after incubation of cells with 20 mM β-cyclodextrin for 4 h. Data of 3 independent (N = 3) experiments performed in quadruplicate are shown as the mean ± S.E.M. Asterisks indicate significant differences based on two-sample *t*-tests (**C**) or two-way ANOVA followed by Tukey’s post hoc test (**A**,**B**). * *p* < 0.05, ** *p* < 0.01, and *** *p* < 0.001.

**Figure 5 cells-11-00639-f005:**
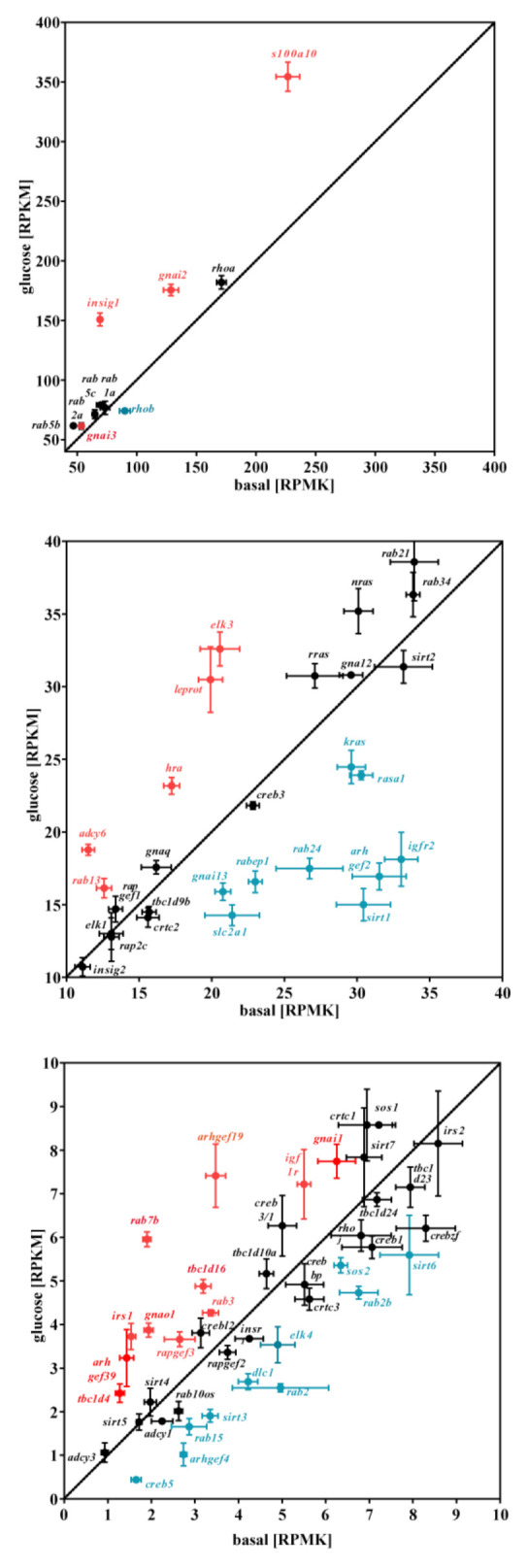
Effects of glucose on mRNA expression of single genes involved in intracellular signaling processes in mHypoA-2/10 cells. mHypoA-2/10 cells were stimulated according to protocol 1 illustrated in Section 2.5 and RPKM determined by RNA-seq. For each condition, triplicates from one single experiment (N = 1) were analyzed as described under experimental procedures. RPKM values for 99 genes are given. Genes marked in red were significantly upregulated, and those marked blue were downregulated by glucose (Bonferroni-corrected *p*-value of ≤0.05). Black genes were not significantly affected by glucose. In the lower panel are genes with RPKM between 1 and 10, in the middle panel between 10 and 40, and in the upper panel between 40 and 400.

**Figure 6 cells-11-00639-f006:**
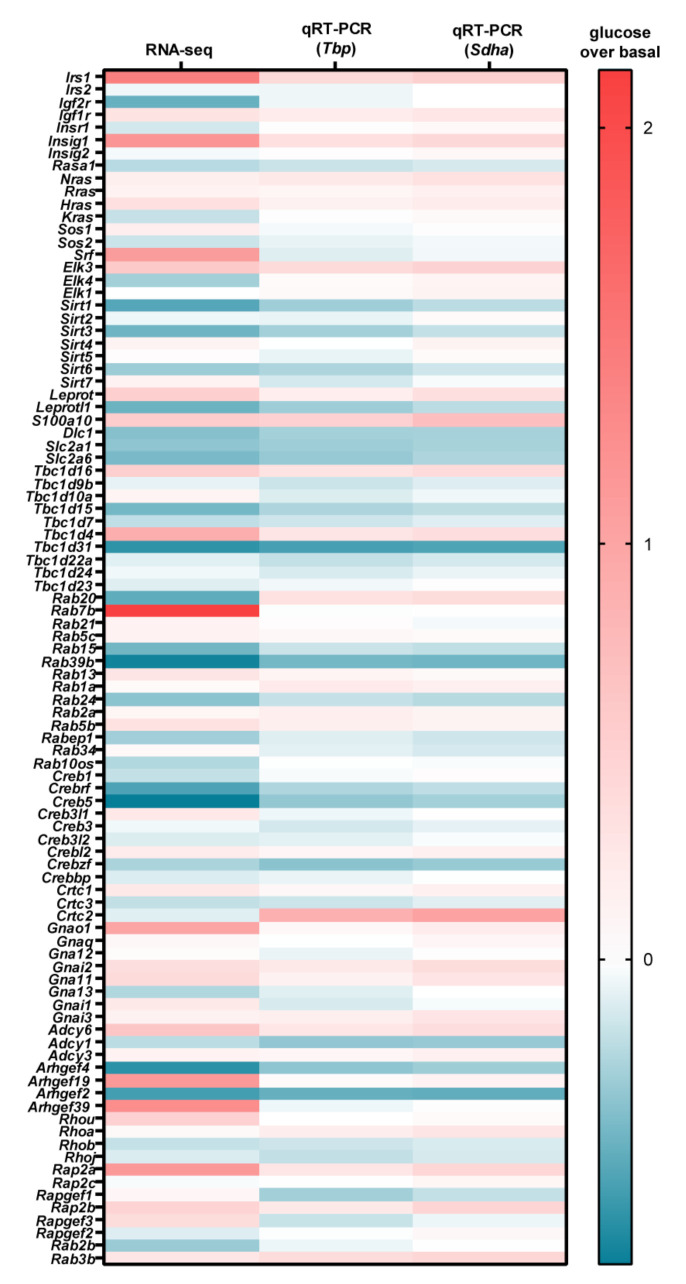
Heat map of glucose-dependent gene expression in mHypoA-2/10 cells combining RNA-seq and qRT-PCR data in mHypoA-2/10 cells. mHypoA-2/10 cells were stimulated accordingly to protocol 1 illustrated in Section 2.5 and RNA levels determined by RNA-seq or qRT-PCR with *Sdha* or *Tbp* as a reference gene. Reddish colors indicate gene induction; blueish colors indicate depression. For qRT-PCR experiments, data of 3 independent (N = 3) experiments performed in triplicate are shown. Data are also provided in Appendix A.

**Figure 7 cells-11-00639-f007:**
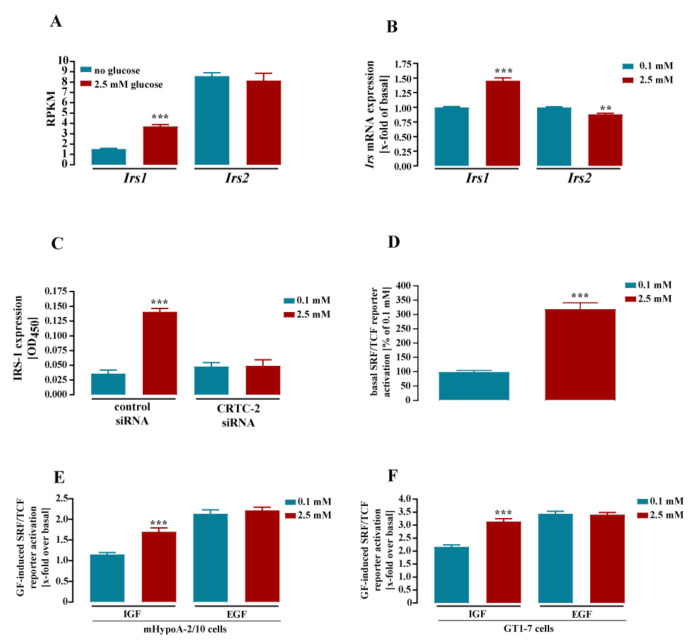
Glucose enhanced IRS-1 expression via CRTC-2 and IGF- but not EGF-induced activation of an SRF/TCF-dependent luciferase reporter gene. mHypoA-2/10 cells were stimulated accordingly to protocol 1 (**A**) or 2 (**B**–**F**) illustrated in Section 2.5. Gene expression was determined by RNA-seq in (**A**) or by qRT-PCR in (**B**). (**A**) For each condition, triplicates were analyzed for Irs1 or -2 expression. (**B**) Irs-1 and -2 expression was analyzed with Sdha as the reference in 3 independent experiments performed in triplicate (mean ± S.E.M.). Data obtained at 0.1 mM glucose were set to 1.0. In (**A**), asterisks indicate significant differences calculated on the basis of Bonferroni-corrected *p*-values of the entire data set obtained by RNA-seq. In (**B**), asterisks indicate significant differences according to two-way ANOVA followed by Tukey’s post hoc test. (**C**) IRS-1 expression was analyzed after transfection of a control or a specific siRNA against CRTC-2, IRS-1 expression by ELISA. Data of 3 independent experiments (N = 3) performed in quadruplicate are shown as the mean ± S.E.M. Asterisks indicate significant differences according to two-way ANOVA followed by Tukey’s post hoc test. (**D**,**E**) Cells were transfected with an SRF/TCF-dependent-reporter gene construct, cultured for 24 h either with 0.1 or with 2.5 mM glucose, stimulated or not with IGF or EGF (both 100 nM) for 4 h and with luciferase activity determined. In (**D**), basal reporter activity is shown normalized to the basal obtained at 0.1 mM (100%). In (**E**), the corresponding basal value was set to 100% and GF-induced reporter activation was calculated as the percentage of basal. Data of 4 independent experiments (N = 4) performed in triplicate are expressed as the mean ± S.E.M. Asterisks indicate significant differences according to two-sampled t-tests (**D**) or two-way ANOVA followed by Tukey’s post-hoc test (**E**). In (**F**), GT1-7 cells were transfected with an SRF/TCF-dependent-reporter gene construct, cultured for 24 either with 0.1 or with 2.5 mM glucose, stimulated or not with IGF or EGF (both 100 nM) for 4 h and with luciferase activity determined. Data of 4 independent experiments (N = 4) performed in triplicate are expressed as the mean ± S.E.M. Asterisks indicate significant differences according to two-way ANOVA followed by Tukey’s post hoc test. ** *p* < 0.01, and *** *p* < 0.001.

**Figure 8 cells-11-00639-f008:**
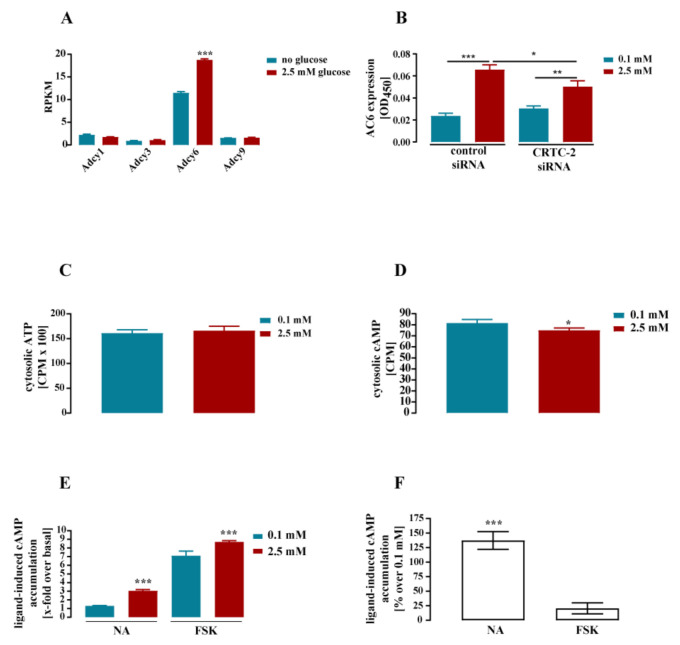
Glucose enhanced AC-6 expression and NA-induced cAMP accumulation in hypothalamic cells. mHypoA-2/10 cells were stimulated accordingly to protocol 1 (**A**) or 2 (**B**–**F**) illustrated in Section 2.5. RPKM were determined by RNA-seq in (**A**). For each condition, triplicates were analyzed as described under experimental procedures. Only genes with at least one RPKM ≥ 1 were considered. Data are expressed as the mean ± S.E.M. Asterisks indicate significant differences calculated on the basis of Bonferroni-corrected *p*-values of the entire data set obtained by RNA-seq. (**B**) After transfection of a control or a specific siRNA against CRTC-2, AC-6 expression was determined by ELISA. Data of 3 independent experiments performed in quadruplicates are shown as the mean ± S.E.M. Asterisks indicate significant differences according to two-way ANOVA followed by Tukey’s post-hoc test. (**C**–**F**) Cells were labeled with [^3^H]-adenine, cultured for 24 h either with 0.1 or with 2.5 mM glucose, stimulated or not with 10 µM FSK or noradenaline (NA) for 20 min, and [^3^H]-cAMP and [^3^H]-ATP were determined. (**C**) Basal [^3^H]-ATP and (**D**) basal [^3^H]-cAMP levels are shown. In (**E**), ligand-induced signals of [^3^H]-cAMP signals are shown as x-fold over basal. In (**F**), ligand-induced signals at 0.1 mM glucose were set to 100%, and ligand-induced signals at 2.5 mM were calculated as the percentage. Data of 4 independent experiments (N = 4) performed in triplicate are expressed as the mean ± S.E.M. Asterisks indicate significant differences according to two-sampled t-tests (**D**,**F**) or two-way ANOVA followed by Tukey’s post hoc test (**E**). * *p* < 0.05, ** *p* < 0.01, and *** *p* < 0.001.

**Table 1 cells-11-00639-t001:** mHypoA-2/10 cells were stimulated according to protocol 1 illustrated in Section 2.5, and RNA levels were determined by RNA-seq or qRT-PCR with *Sdha or Tbp* as a reference gene. For the RNA-seq data, differences were calculated on the basis of Bonferroni-corrected *p*-values. For the qRT-PCR data, ΔΔ*Cp* using Sdha or Tbp was calculated, and significant differences were determined using two-way ANOVA and Sidak’s post-test or two-sample t-tests. Mean change indicates alterations based on all three data sets. White background indicates gene induction, grey background gene depression.

Gene		RNA-seq		qRT-PCR						
Name	Function	Change (%)	Bon-ferroni-Corrected	tbp			Sdha			Mean Change (%)
				Change (%)	Sidak’s ΔΔCp ANOVA	ΔΔCp *t*-Test	Change (%)	Sidak’s ΔΔCp ANOVA	ΔΔCp *t*-Test	
*Irs1*	adaptor protein for IR and IGFR	123	***	41	***	*	52	***	*	72 ± 26
*Igfr1*	IGF receptor	34	***	21	**	***	28	**	***	28 ± 4
*Adcy6*	cAMP production	66	***	28	**	***	37	**	**	44 ± 11
*Gnai1*	inhibition of cAMP production	43	***	15	*	***	30	*	**	29 ± 8
*Gnai2*	inhibition of cAMP production	40	***	24	**	***	39	***	***	35 ± 5
*Gnai3*	inhibition of cAMP production	16	*	17	*	**	30	*	***	21 ± 5
*Gna11*	PLC activation/Ca^2+^	17	*	24	*	**	30	*	**	24 ± 4
*Hras*	monomeric G protein/MAPK activation	38	***	15	*	***	22	*	**	25 ± 7
*Elk3*	transriptional activator/SRF/SRE	61	***	40	***	**	50	***	**	50 ± 6
*Insig1*	cholesterolsynthesis inhibitor	108	***	34	***	**	43	***	**	62 ± 23
*Rap2a*	monomeric G protein/MAPK activation	105	***	27	***	**	45	***	**	59 ± 24
*Tbc1d1*	Rab inhibitor	90	***	29	***	**	38	***	**	52 ± 19
*S100a10*	calcium-binding protein	59	***	51	***	**	72	***	**	61 ± 6
*Arhgef2*	Rho activator	−58	***	−46	***	**	−47	***	***	−50 ± 4
*Tbc1d31*	Rab inhibitor	−61	***	−55	***	***	−53	***	***	−56 ± 2

## Data Availability

The data presented in this study are available on request from the corresponding author. Raw RNA-seq data are available under ArrayExpress accession E-MTAB-10629.

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
