# Peer review of "Analysis of the Glucose-Dependent Transcriptome in Murine Hypothalamic Cells"

_cells, 2022, doi:10.3390/cells11040639_

Round 1

Reviewer 1 Report

The paper contains useful data but needs some corrections.

1) First, there is a problem with interpretation of experimental conditions. The authors consider starving conditions with 0.1 mM of glucose as a basal while the 2.5 mM as high glucose level. 
However, this classification confuse conclusions because basal level should be defined as normoglycemic. In vivo experiments performed in rodents (Silver and Erecinska 1994; Dunn-Meynell et al., 2009; Mayer et al., 2006) showed that the brain level of glucose in normoglycemic conditions is in the range of about 1.34 to 2.4 mM (mean) +/- of about 0.1 mM. Therefore, the “high level” of 2.5 mM is in fact in the range observed in basal conditions. In turn, the “basal level”  of 0.1 mM is in fact in the range of glucose levels observed in severe hypoglycemic conditions. Therefore, observed transcriptomic effects are not resulting from high level of glucose but are caused by shift from sever hypoglycemia to basal normoglycemic conditions. Authors should correct their classification of experimental conditions because in the current version they lead to confusion. 

2) The complete list of significantly altered genes in RNA-seq data should be provided as an Excel supplementary file together with fold changes, p values and corrected p values. This will greatly increase the impact of the study. 

3) The authors should provide information about the depth of sequencing and number of sequenced samples.

4) The symbols of genes should be provided in correct  format including organism-specific formatting guidelines (please see for example https://www.biosciencewriters.com/Guidelines-for-Formatting-Gene-and-Protein-Names.aspx).

5) In two places there is an erroneous gene symbol CRCT-2.

6) There is a confusion about the method used to correct data for multiple comparisons because two different methods are provided in the method section while only one is mentioned in result section. 
It should be noted that Benjamini and Hochberg method is more appropriate for transcriptomic data. 
Furthermore, FDR is not the “false positive rate” and Bonferroni correction is not a FDR-controlling procedure (it is controlling the familywise error rate – it is a different concept).

7) What is the background of the cell lines and their phenotype? They are neuronal or astrocytic-like cells? They were obtained from mice? Please, specify it. This is an important information for readers.

8) Please, provide number of samples for all groups. Such an information can be included in figure legends. 

9) Why the authors used both the student’s t-test and one‐way ANOVA? They are equivalent. 

Author Response

Please see attched file

Reviewer 2 Report

In the present paper, Webert and colleagues analysed the glucose-dependent transcriptome of a neuronal hypothalamic cell line (the mHypoA-2/10 cells) by total RNA-seq, and qRT-PCR for 100 selected genes. They show that 831 genes were up-regulated by glucose whereas 1390 were down-regulated. They notably found that key genes involved in the cholesterol biosynthesis pathway were up-regulated by glucose. They confirmed these data in another hypothalamic neuronal cell line. They conclude that glucose impacts lipid synthesis and some intracellular signaling processes in hypothalamic neurons and speculate on the consequences of this impact in pathophysiological condition of chronic high glucose levels.

The paper is well written and organized, and the various experiments are clearly exposed and seem seriously performed. However, this paper shows important limitation and a number of points deserve clarification:

  • The authors should justify the choice of the mHypoA-2/10 cell line among all the mHypoA cell lines that are available and why it is a pertinent choice.
  • They should confirm the physiological relevance of their findings by using a better physiological model such as primary culture of hypothalamic neurons or hypothalamic explants.
  • The authors should also clarify their protocol for glucose stimulation of the cells. They incubate the cells with 0.1 mM or 2.5 mM glucose for 3 hours before their RNA seq analysis. However, the cells before this treatment were cultured in DNEM containing 25 mM glucose. This means that the mHypoA cells were chronically submitted to a very high dose of glucose before being tested with very low doses of glucose. How the authors can be sure that this chronic treatment of the cells with very high concentrations of glucose does not impact the expression of the genes analyzed? In addition, the physiological concentration of glucose in the brain is 2 mM, so when the authors stimulate the cells with 2.5 mM of glucose, they do not stimulate physiologically the cells, it corresponds rather to basal conditions for the hypothalamic neurons. A stimulation with 5 mM of glucose to stimulate the neurons is commonly reported in the literature. Conversely, 0.1 mM of glucose does not reflect basal condition but very low glucose concentrations.
  • In the discussion, the authors try to link their data to pathophysiological conditions, i.e. diabetes. It is too speculative! However, it would have been of great interest to perform the same RNA-seq analysis on cells chronically treated with high glucose concentrations and on cells treated with palmitate to mimick the lipotoxic conditions of diabetes.

Minor comments:

  • Fig. 3A: the statistical analysis between basal and glucose is missing in CRTC-2 siRNA.
  • The resolution of figs. 4 and 5 is bad. Impossible to read what is written. Please correct this problem.

Author Response

Please see attched file

Reviewer 3 Report

Current manuscript (Leonhard Webert et al) describing the consequences in hypothalamic gene expression and signaling in response to extracellular glucose levels. Based on the in vitro cell culture (murine hypothalamic cells-mHypoA-2/10) model, authors performed the RNA transcriptome analysis in response to high glucose, found differential gene expression patterns (Up or down regulated) and most notable pathways identified is cholesterol biosynthesis and intracellular signaling genes related to cAMP production. These findings are important to understand the physiological effect of extracellular glucose alone or together with free fatty acid and higher cholesterol levels which mimic the patients with diabetes condition. In addition, these differential regulation in signaling pathways can contribute to pathological condition and disease progression in Diabetes.  However, several major and minor concerns related to current study need to address by authors.   

Major comments:

  • Under normal physiological condition in patients with Diabetes develop high glucose together with high free fatty acid (TG) and cholesterol. However, Current manuscript all the in vitro cell culture studies performed under low or high glucose condition. In Patients with Diabetes, Hypothalamic region of Brain contain only high glucose alone or together with TG and cholesterol, can authors comment about this?
  • Authors need to show the figure related to pathway analysis (List of pathways up or down regulated) indicating that the most significant pathway identified and regulated is cholesterol biosynthesis together with other regulated pathways (example: mitochondrial oxosphos, glycolysis, apoptosis, intracellular GPCR signaling) involved. This will allow to clearly identify the affected pathways under high glucose condition.  
  • In Figure 2, authors showed regulation of Fatty acid (Srebp1) and Cholesterol (Srebp2) biosynthesis in response to high glucose in mHypoA-2/10 cells. Results showing Srebp1 is increased significantly under high glucose. However, authors did show expression of target genes related to Srebp1 pathway. Need to show all the genes related to Srebp1 pathway.
  • In Figure 3A, authors showed the protein expression of SREBP1 and SREBP2. As of my knowledge, SREBP1 and SREBP2 regulated Post translationally. Therefore, measurement of SREBP expression levels by ELISA is not valid and also source is not validated. SREBPs protein expression levels need to show by westernblot analysis only.
  • In Figure 2, 3 and 6, Bar graphs indicating that cells were culture in Basal glucose or 0.1 mM Glucose. What is concentration of Glucose under Basal? Authors need to represent all the Labels consistently.

Minor comments:

  • Sections 2.2, methods part, source of cells (catalog et al) need to mention clearly. Also, what is GT1-7 cells? In which experimental part these cells has been used? I do not see any results part that is generated from these GT1-7 cells.
  • Sections 2.5, authors mentioned cells kept at 0 mM. However, in the several parts of manuscript authors mentioned that they used 0.1 mM Glucose in experiments. Need to clarify clearly.
  • Section 2.7 Whole Cell ELISA, what is Source of antibodies like catalog and company details et.
  • Abbreviate the RPKM in Figure 2.
  • In Figure 5, SDHA and TBP used as hpouse keeping genes. Have to abbreviate the Gene full name.

Author Response

Reviewer #3

Comments and Suggestions for Authors

Current manuscript (Leonhard Webert et al) describing the consequences in hypothalamic gene expression and signaling in response to extracellular glucose levels. Based on the in vitro cell culture (murine hypothalamic cells-mHypoA-2/10) model, authors performed the RNA transcriptome analysis in response to high glucose, found differential gene expression patterns (Up or down regulated) and most notable pathways identified is cholesterol biosynthesis and intracellular signaling genes related to cAMP production. These findings are important to understand the physiological effect of extracellular glucose alone or together with free fatty acid and higher cholesterol levels which mimic the patients with diabetes condition. In addition, these differential regulation in signaling pathways can contribute to pathological condition and disease progression in Diabetes.  However, several major and minor concerns related to current study need to address by authors.   

Major comments:

  • Under normal physiological condition in patients with Diabetes develop high glucose together with high free fatty acid (TG) and cholesterol. However, Current manuscript all the in vitro cell culture studies performed under low or high glucose condition. In Patients with Diabetes, Hypothalamic region of Brain contain only high glucose alone or together with TG and cholesterol, can authors comment about this?

The goal of the present manuscript is to monitor the overall effects of changes in extracellular glucose concentrations on the gene expression of hypothalamic cells. Although these data are also of high interest for Diabetes research, our goal was not to mimic diabetic conditions in cell culture. However, given that glucose enhanced cholesterol levels in the supernatant of the cells, apparently our treatment indirectly also affected this aspect of diabetic conditions. To further dissect whether the observed effects on gene expression are primarily caused by glucose or secondarily by fatty acids or cholesterol or a combination of both, will be an interesting task for future studies.

  • Authors need to show the figure related to pathway analysis (List of pathways up or down regulated) indicating that the most significant pathway identified and regulated is cholesterol biosynthesis together with other regulated pathways (example: mitochondrial oxosphos, glycolysis, apoptosis, intracellular GPCR signaling) involved. This will allow to clearly identify the affected pathways under high glucose condition.  

We thought this is a great idea and added such a figure to the revised version of the manuscript (Fig. 2).

3) In Figure 2, authors showed regulation of Fatty acid (Srebp1) and Cholesterol (Srebp2) biosynthesis in response to high glucose in mHypoA-2/10 cells. Results showing Srebp1 is increased significantly under high glucose. However, authors did show expression of target genes related to Srebp1 pathway. Need to show all the genes related to Srebp1 pathway.

We are grateful for this suggestion but, we are not sure to which “SREPB1 pathway” the reviewer refers to. However, we added expression data of ATP citrate lyase, acetly-Coa carboxylase-α, fatty acid synthase and stearoyl-Coa desaturase to the new version of the manuscript (Fig. 3 B). Interesting, all of these genes were indeed upregulated by glucose.

  • In Figure 3A, authors showed the protein expression of SREBP1 and SREBP2. As of my knowledge, SREBP1 and SREBP2 regulated Post translationally. Therefore, measurement of SREBP expression levels by ELISA is not valid and also source is not validated. SREBPs protein expression levels need to show by westernblot analysis only.

We do agree that SREBP proteins are regulated on the posttranslational level but, we do not see why such modifications would affect protein detection by ELISA and not by western-blot. In contrast, when analyzed by western-blot differently modified protein pools will migrate at distinct heights making the quantification of the single bands even more complicated. Further, ELISA approaches are faster, easier to quantify and more efficient, since more samples can be tested within a shorter timeframe. Because we did not only analyze effects of glucose on SREBPs expression alone but, also aimed at simultaneously monitoring effects of siRNAs against CRTC-2 on SREBPs expression, the number of samples is just too high for western-blot analysis.

  • In Figure 2, 3 and 6, Bar graphs indicating that cells were culture in Basal glucose or 0.1 mM Glucose. What is concentration of Glucose under Basal? Authors need to represent all the Labels consistently.

As mentioned in section 2.5 of the manuscript, we used two different protocols for glucose stimulation. These protocols were recently established from us (Breit A, Mol Endocrinol. 2016 Jul; 30 (7):748-62). In deed the basal levels are distinct: in protocol 1, “basal” was determined in cells cultured for 24 h with 0.1 mM glucose and then for 3 h with no glucose. In protocol 2, “basal” was determined in cells cultured for 24 h in 0.1 mM glucose. Data of cells shown in figures 2,3 and 6 A&B (old manuscript, now figures 3,4 and 7 A&B) were stimulated accordingly to protocol 1. We agree that this is confusing for the reader. Thus, we added a cartoon to section 2.5 indicating the stimulation times and glucose concentrations for both protocols, we point to this cartoon in the figure legends and changed the labeling of the figures.

Minor comments:

  • Sections 2.2, methods part, source of cells (catalog et al) need to mention clearly.

We point to the fact that CLU-176 is the ordering number of Cedarlane for mHypoA-2/10 cells.

Also, what is GT1-7 cells? In which experimental part these cells has been used? I do not see any results part that is generated from these GT1-7 cells.

Data shown in the old figures 3C (now 4C) and 6C (now 7 C) were generated with GT1-7 cells. This is indicated in the labeling of the y-axes, in the figure legend and in the case of figure 7 C also directly in the figure. In the revised version of the manuscript, we also provide two additional citation for GT1-7 cells in section 2.2.

  • Sections 2.5, authors mentioned cells kept at 0 mM. However, in the several parts of manuscript authors mentioned that they used 0.1 mM Glucose in experiments. Need to clarify clearly.

Please see major point 4.

  • Section 2.7 Whole Cell ELISA, what is Source of antibodies like catalog and company details et.

This information is provided in the section 2.1 “materials”.

  • Abbreviate the RPKM in Figure 2.

            This is not clear to us: RPKM is already an abbreviation.

  • In Figure 5, SDHA and TBP used as hpouse keeping genes. Have to abbreviate the Gene full name.

            This is not clear to us: Sdha and Tbp are already abbreviations

Round 2

Reviewer 2 Report

The authors have answered appropriately and honestly to most of my questions/criticisms.

The paper is substancially improved in its revised form.

Author Response

We are thankful to the reviewer for her/his useful comment that helped us improving the manuscript.

Reviewer 3 Report

All the comments were addressed properly. However, authors need to add information regarding fallowing minor and important concern. 

  • In methods 2.7, Line 174: authors need to verify/add the source of the Srebp-2 and Srebp-2 antibodies. It is important to know the readers regarding materials used by authors. 

Author Response

As requested, we added the source and catalog number of the used antibodies to section 2.7.